# Fabrication of Metal-Insulator-Metal Nanostructures Composed of Au-MgF_2_-Au and Its Potential in Responding to Two Different Factors in Sample Solutions Using Individual Plasmon Modes

**DOI:** 10.3390/mi13020257

**Published:** 2022-02-03

**Authors:** Hirotaka Yamada, Daiki Kawasaki, Kenji Sueyoshi, Hideaki Hisamoto, Tatsuro Endo

**Affiliations:** 1Department of Applied Chemistry, Graduate School of Engineering, Osaka Prefecture University, Sakai 599-8531, Japan; sxb02146@edu.osakafu-u.ac.jp (H.Y.); syb02029@edu.osakafu-u.ac.jp (D.K.); sueyoshi@chem.osakafu-u.ac.jp (K.S.); hisamoto@chem.osakafu-u.ac.jp (H.H.); 2Precursory Research for Embryonic Science and Technology (PRESTO), Japan Science and Technology Agency (JST), 5-3 Yonban-cho, Chiyoda, Tokyo 102-8666, Japan

**Keywords:** plasmonics, metal-insulator-metal, plasmonic sensor, dual-plasmon modes

## Abstract

In this paper, metal–insulator–metal (MIM) nanostructures, which were designed to exhibit two absorption peaks within 500–1100 nm wavelength range, were fabricated using magnesium difluoride (MgF_2_) as the insulator layer. Since the MIM nanostructures have two plasmon modes corresponding to the absorption peaks, they independently responded to the changes in two phases: the surrounding medium and the inside insulator layer, the structure is expected to obtain multiple information from sample solution: refractive index (RI) and molecular interaction between solution components and the insulator layer. The fabricated MIM nanostructure had a diameter of 139.6 ± 2.8 nm and a slope of 70°, and exhibited absorption peaks derived from individual plasmon modes at the 719 and 907 nm wavelengths. The evaluation of the response to surrounding solution component of the MIM nanostructures revealed a linear response of one plasmon mode toward the RI of the surrounding medium and a large blue shift of the other plasmon mode under conditions where glycerol was present at high concentration. From optical simulation and the evaluation of the MgF_2_ fabricated by deposition, the blue shift was expected to be due to the swelling of MgF_2_ interacting with the hydroxyl groups abundantly included in the glycerol molecules. The results indicated the individual responses of two plasmon modes in MIM nanostructures toward medium components, and brought the prospect for the simultaneous measurement of multiple elements using two or more plasmon modes.

## 1. Introduction

Metal nanostructures have been studied for highly efficient optical materials and biochemical sensor applications because of the unique optical characteristics derived from the collective oscillation of free electrons on their surfaces, i.e., the localized surface plasmon (LSP) [1,2,3,4,5,6]. LSP characteristics, such as the absorption of light and the generation of electric fields around metal nanostructures, strongly depend on the shapes of the metal nanostructures and the surrounding dielectric environment (refractive index, RI) [7,8,9]. Therefore, metal nanostructures have been studied for their applications in highly sensitive, label-free sensors that detect surrounding RI changes caused by adsorption or molecule interaction on metal nanostructure surfaces [10,11]. Because the RI sensitivity depends on the properties of the LSP mode, various shapes of metal nanostructures have been studied for improving sensitivity [12,13,14]. In the field of RI sensing technology, through improvements in fabrication [15,16,17], numerous studies on the label-free and real-time detection of several targets such as cells [18,19,20], DNA [21,22,23], antigens [24,25,26], and ions [27,28] have been reported.

These LSP-based sensors are generally composed of devices with metal nanostructures and a responsive layer (e.g., antibody and single-strand DNA), and detect target molecules by measuring the wavelength shift of single absorption peak derived from the LSP. Consequently, conventional LSP-based sensors have been used to detect single parameters, such as the concentration of a single kind of molecule. Therefore, it is challenging to develop an LSP-based RI sensing device that can detect the concentration and components of more than one parameter in a sample. This problem limits the application range of LSP-based sensors, especially in the case of measuring the correlation of multiple data from the surrounding medium is required, such as, for example, investigations on the biochemical behavior of living cells [29,30,31].

To overcome this challenge, we focused on metal-insulator-metal (MIM) nanostructures. MIM nanostructures are composed of multi-layered nanostructures in which two metal layers are separated by an insulator layer, and derive optical properties from the propagating surface plasmon (PSP)-PSP coupling [32,33] LSP-LSP interactions between two metal layers [34,35,36]. In LSP-LSP interacting conditions, the LSP mode is divided into a higher energy state and a lower energy state, resulting in multi-modal LSP properties. In addition, since the multiple LSP modes formed through LSP-LSP coupling derive electric fields in spatially different locations at different wavelengths, MIM nanostructures are attracting attention as highly functional optical elements. Therefore, MIM nanostructures are expected to be applied to obtain multiple kinds information from a sample by utilizing the property of forming electric fields at different spatial locations, in combination with spatial material distribution. However, previously reported MIM nanostructures used SiO_2_, which has high chemical stability, as the insulator layer, the LSP mode which was reported to form an electric field in the dielectric layer formed by the LSP-LSP interaction was not used for detection [37,38,39]. In other words, the multi-mode properties of MIM nanostructures have never been applied to measurements.

In this study, we investigated the simultaneous response property of disk-stacked MIM nanostructures composed of magnesium fluoride (MgF_2_) as the insulator layer and gold (Au) as the metal layers. The disk-stacked MIM nanostructures are expected to have two LSP modes that generate electric fields outside the MIM nanostructures and inside the insulator layer, and are expected to independently respond to the changes in two phases: the surrounding medium and inside the insulator layer. In addition, by introducing MgF_2_, which is known to adsorb hydroxyl groups of water molecules [40,41] as the insulator layer, the MIM nanostructures are expected to respond to the hydroxyl group by the LSP mode, which generates electric fields inside the insulator layer. By combining the individual mode distribution and the adsorption characteristics of MgF_2_, the MIM nanostructures are expected to monitor the information of both the RI change in the solution and the changes within the MgF_2_ layer (corresponding to the interaction of molecules in solution with MgF_2_).

First, the MIM nanostructures that have LSP modes that generate electric fields in spatially different locations at different wavelengths were designed. Based on the design, MIM nanostructures were fabricated using electron-beam lithography, and their structural and optical characteristics were evaluated by observing scanning electron microscopy (SEM) images and measuring the absorption spectrum. Finally, to investigate the response of MIM nanostructures to the RI value and molecular components of a surrounding sample solution, the absorption spectrum was evaluated using solutions with several components—water, isopropyl alcohol (IPA), and glycerol—that have different numbers of hydroxyl groups and are expected to have different adsorption reactions with MgF_2_.

## 2. Materials and Methods

### 2.1. Materials

A 500 μm-thickness glass substrate purchased from Matsunami Glass Ind., Ltd. (Osaka, Japan) was used for the substrate supporting the MIM nanostructures. MgF_2_ purchased from Kojundo Chemical Lab. Co., Ltd. (Saitama, Japan) and Au purchased from Tanaka Kikinzoku Co., Ltd. (Tokyo, Japan) were used as the materials of the MIM nanostructures. Hexamethyldisilazane purchased from Zeon Co. (Tokyo, Japan) was used for the hydrophobization of the glass substrate. Anisole was purchased from FUJIFILM Wako Pure Chemical Industries, Ltd. (Osaka, Japan) to dilute and remove positive-tone the electron-beam resist ZEP520A, which was purchased from Zeon Co. (Tokyo, Japan). A conductive polymer, ESpaser 300Z, purchased from Showa Denko K. K. (Tokyo, Japan), was used to prevent the accumulation of electrons during electron-beam exposure. ZED-N50 purchased from Zeon Co. (Tokyo, Japan) was used as developer of ZEP520A after electron-beam exposure. Isopropanol (IPA) purchased from Kanto Chemical Co. (Tokyo, Japan) and glycerol purchased from FUJIFILM Wako Pure Chemical Industries, Ltd. (Osaka, Japan) were used to prepare sample solutions with different refractive index values and components.

### 2.2. Methods

#### 2.2.1. Design of MIM Nanostrctures

The MIM nanostructures were designed to have two individual absorption peaks at 600–1000 nm wavelength region, in which the optical measurement could be carried out by a single conventional spectrometer. The structural design was evaluated using the optical simulation software Lumerical FDTD (Lumerical Solutions, Inc., Vancouver, BC, Canada).

In the simulation analysis, an Au-MgF_2_-Au MIM nanostructure was modelled on SiO_2_ substrate, and the light was introduced from the substrate. By comparing the transmitted light through the substrate with and without the MIM nanostructure, the absorption spectrum was obtained. The optical characteristics of Au and SiO_2_ were fitted to the material databases included in the simulation software, based on the CRC Handbook of Chemistry and Physics and the Handbook of Optical Constants of Solids I–III by E. Palik, respectively. The MgF_2_ was modelled to have its RI constant at 1.366 in every wavelength, considering the reduction of RI through the deposition process in the experiment.

In addition to the absorption spectrum of the designed MIM nanostructures, the electric field distributions around the nanostructure were evaluated to clarify the LSP mode characteristics at each peak wavelength.

#### 2.2.2. Fabrication of MIM Nanostructures

A schematic illustration of the fabrication process of the MIM nanostructures is shown in Figure 1. Here, to fabricate the square array of MIM nanostructure with 140 nm of diameter, electron-beam lithography was performed. [42,43]. First, positive tone resist ZEP520A was diluted to 1.5 times by anisole. Then, hexamethyldisilazane, the positive-tone resist ZEP520A, and conductive polymer ESpacer 300Z [44,45] were spin-coated on a glass substrate for 60 s at 3000, 4000, and 3000 rpm, respectively. After each spin-coating step, the substrate was baked at 180, 180, and 85 °C, respectively. The substrate was then exposed to an electron-beam. The substrate was washed in flowing water to remove the ESpacer 300Z and dried with an air blower. The substrate was then developed using ZED-N50 for 1.5 min at 25 °C and dried.

The bottom 30 nm thickness of Au, center 20 nm thickness of MgF_2_, and top 30 nm thickness of Au were deposited in this order using a thermal evaporator (SVC-700TM, Sanyu Electron Co., Ltd., Tokyo, Japan) on the substrate with hole-patterned resist. Between each layer, platinum approximately 5 nm thick was sputtered as an adhesion layer. Finally, lift-off was performed by anisole to remove the resist layer, and the substrate with MIM nanostructures was obtained. The diameter and layered structure of the MIM nanostructures were evaluated using a field-emission scanning electron microscope (FE-SEM) system (JSM-7610F, JEOL Ltd., Tokyo, Japan).

#### 2.2.3. Evaluation of Optical Characteristics and Refractive Index Response of MIM Nanostructures

To evaluate the LSP property of the fabricated MIM nanostructures, the absorption spectrum was observed using the optical micro-spectroscopy system shown in Figure 2. First, white light was irradiated from a tungsten light source (SLS201L/M, Thorlabs, Newton, NJ, USA) and introduced to the bottom objective lens (20×/NA 0.45) via two mirrors. The spot size on the substrate approximated 100 μm, and the transmitted light was captured by a top objective lens (10×/NA 0.28). Finally, the transmitted light was divided by a half mirror, diverting it between the ocular lens and the detector, a spectrum analyzer (AQ6370D, Yokogawa Electric Co., Tokyo, Japan).

The absorption intensity at each wavelength was calculated by the following equation. Here, *I*_abs_ is the absorption intensity, *I*_trans,plas_ is the transmission intensity through the MIM nanostructures and *I*_trans,glass_ is the transmission intensity through the glass substrate.
*I*_ext_ = 1 − *I*_refl,plas_/*I*_refl,flat_(1)

Since the light at a specific wavelength is absorbed through LSPR, the plasmon modes can be found as the absorption peaks of the absorption spectrum. To clarify the absorption peaks from the absorption spectrum, the absorption spectrum was fitted assuming that the spectrum was composed of two Lorentzian functions. In the fitting step, the difference between the sum of the two Lorentzian functions and the experimentally obtained spectrum was minimized in the wavelength range of 600 to 1000 nm, in which the spectral noise and absorption of Au could be ignored.

The responsivities of the MIM nanostructures to the surrounding medium were evaluated using sample solutions with different RI values and components: 0–100 wt.% isopropyl alcohol (IPA) aqueous solution (Water-IPA); 0–32 wt.% glycerol aqueous solution (Water-Gly); and 0–32 wt.% glycerol-IPA mixture solution (IPA-Gly). The RI values of each solution were measured to be 1.3324–1.3753 for Water-IPA, 1.3324–1.3690 for Water-Gly, and 1.3753–1.4009 for IPA-Gly, using a hand-held refractometer (PAL-RI, ATAGO Co., Ltd., Tokyo, Japan). Table 1 shows the detailed RI values of each solution. The absorption peaks of the binding and anti-binding modes were measured for each solution, and the response of the peaks was evaluated.

In the case where the absorption peak responded to the RI value of the sample solution linearly, the responsivities were evaluated from the slope of the refractive index-absorption peak shift plots. Here, *λ* is the absorption peak wavelength and *n* is the refractive index.
Responsivity [nm/RIU] = ∂*λ*/∂*n*(2)

## 3. Results

### 3.1. Simulated Optical Characteristics of Designed MIM Nanostructures

The structures and absorption peaks of the MIM nanostructures designed in this work are shown in Figure 3a,b. Here, the MIM nanostructures were designed to have two individual absorption peaks in the 600–1000 nm wavelength region in the wet condition (surrounding refractive index: 1.333). From the absorption spectrum, MIM nanostructures which have absorption peaks at 715 and 920 nm wavelength were successfully designed by setting its their parameters as follows: 30 nm Au layer thickness; 20 nm MgF_2_ layer thickness; and 140 nm diameter.

Figure 4a,b present the simulated electric field distributions at the 715 and 920 nm wavelengths, which are the absorption peaks in Figure 3b, respectively. From these results, the electric field was generated surrounding MIM nanostructures at the 715 nm wavelength. Alternatively, the electric field was generated inside the MIM nanostructures, especially inside the MgF_2_ layer, at the 920 nm wavelength. These results indicated the MIM nanostructures’ design had two plasmon modes expected to independently respond to the changes in two material phases: the surrounding medium and inside the insulator layer. For the detailed investigation of LSP mode characteristics deriving the electric field distributions, the vector component of the electric field was evaluated.

Figure 4c,d show the simulated x-vector component (in the horizonal direction in the figure) of the electric field distributions at 715 and 920 nm wavelength, respectively. From the x-vector component of the electric field distribution at the 715 nm wavelength, the direction of the electric field around top and bottom Au layer was observed to be the same, and the LSP modes of each Au layer were expected to oscillate in the same oscillation phase. Hence, the charges in each Au layer acted repulsively, and, as expected, resulted in electric field generation around the MIM nanostructures. Alternatively, from the x-vector component of electric field distribution at the 920 nm wavelength, the direction of the electric fields around top and bottom Au layer were observed to be opposite, and the LSP modes at each Au layer were expected to oscillate in the opposite oscillation phase. Hence, the charges of each Au layer acted attractively, and, as expected, resulted in electric field generation inside the MgF_2_ layer. From these results, the designed MIM nanostructures were expected to have two plasmon modes formed by the interaction between two LSP modes, from which were derived the electric fields outside and inside the nanostructure at different wavelengths.

### 3.2. Structural and Optical Characterization of Fabricated MIM Nanostrctures

MIM nanostructures were fabricated based on the design described in Section 3.1. Figure 5a,b present the SEM images of the fabricated MIM nanostructures observed from the top and the side. From the SEM image observed from the top, the diameters of the fabricated MIM nanostructures were measured to be 139.6 ± 2.8 nm, and from the SEM image observed from the side, a multi-layered structure was observed. In addition, the SEM image observed from the side showed the tapered structure of fabricated MIM nanostructure, with about 70° of slope.

Figure 6a presents the absorption spectrum of fabricated the MIM nanostructures, and the simulated absorption spectrum of MIM nanostructures imitated the SEM images (140 nm of bottom diameter and 70° of slope). From the simulated spectrum, two peaks were clearly observed at the 694 and 828 nm wavelengths. From the experimental spectrum, the relatively unclear peaks were expected to configure the spectrum. Figure 6b presents the experimental absorption spectrum and two Lorentz functions fitted to the spectrum. From the results, the experimental absorption spectrum was revealed to be composed of two absorption peaks at the 719 and 907 nm wavelengths. Compared to the simulated spectrum, both peaks were observed at longer wavelengths in the experimental spectrum. The difference is expected to due to the fabrication accuracy, especially that of thickness in each layer, and the difference in optical properties of the materials that were actually fabricated and introduced in the simulation. However the difference of the peak wavelength was observed, considering the similarity of structures and the number of peaks in the wavelength region, the peaks in the simulated spectrum and the experimental spectrum were expected to be related. Therefore, to clarify the assignment of the peaks in the experimental spectrum, the electric field distributions at the 694 and 828 nm wavelengths of the simulated MIM nanostructures were evaluated.

Figure 7 shows that the simulated electric field distribution of the MIM nanostructures imitated the SEM images at each peak wavelength. Similar to the designed MIM nanostructures in Section 3.1, the peak at the shorter wavelength (694 nm wavelength) was assigned to the plasmon mode that generated the electric field outside the MIM nanostructures, and the peak at the longer wavelength (828 nm wavelength) was assigned to the plasmon mode that generated the electric field the inside MgF_2_ layer. Therefore, assignment of two absorption peaks observed from MIM nanostructures could be judged from the peak position even when the structure was tapered. From these results, the absorption peaks observed from the fabricated MIM nanostructures were suggested to be assigned to the simulated MIM nanostructures imitating the SEM images, and this suggestion was applied in following studies and discussions.

### 3.3. Surrounding Refractive Index Response of MIM Nanostructures to Sample Solutions with Different Components

Figure 8a presents the varying absorption peak wavelengths of the peak at the shorter wavelength of fabricated MIM nanostructures depending on the RI values of the surrounding sample solutions. In the figure, the gray hollow diamonds indicate the plots for 0–32 wt.% glycerol aqueous solution (Water-Gly), the blue hollow triangles indicate the plots for 0–32 wt.% glycerol-IPA mixture solution (IPA-Gly), and the red hollow squares indicate the plots for 0–100 wt.% IPA aqueous solution (Water-IPA). From the results, the peak at the shorter wavelength shifted to longer wavelengths systematically. Therefore, the plasmon mode assigned to the peak at the shorter wavelength is expected to respond to surrounding RI values, and this result coincides with the peak assignment discussed in Section 3.2 (the peak at the shorter wavelength was assigned to the plasmon mode that generated the electric field outside the MIM nanostructures).

In this result, assuming the RI response is linear, the plots for glycerol concentrations of 24% and 32% (the gray dashed ellipses in Figure 8a) deviated slightly (R^2^ = 0.8926, including the plots; R^2^ = 0.9759, excluding the plots), which may be because of the RI increase in the vicinity of the nanostructure caused by the adsorption of glycerol. The sensitivity to RI was measured to be 259 nm/RIU, excluding the plots for glycerol concentrations of 24% and 32%.

Figure 8b presents the varying absorption peak wavelengths of the peak at the longer wavelength of fabricated MIM nanostructures depending on the RI values of the surrounding sample solutions. In contrast to Figure 8a, the peak at the shorter wavelength showed a non-systematic response to the RI value, which differed according to the components of the solutions. In the case of Water-IPA, the peak wavelength of the binding mode showed a relatively small change compared to the other two cases, and in the cases of Water-Gly and IPA-Gly, a similar response behavior was observed: the peak shifted to a longer wavelength once the surrounding RI increased, and then largely shifted to a shorter wavelength (reaching to 20 nm shift compared to the water: *n* = 1.3324). However, the longer wavelength shift the surrounding RI increased can be described as the RI response of the peak, the large shorter wavelength shift cannot have been due to the surrounding RI change. Therefore, to understand the response background, the possible structural changes that would cause a 20 nm shorter wavelength shift were demonstrated by simulation analysis.

Figure 9a–d illustrate the base MIM nanostructure (30 nm of Au layer thickness; 20 nm of MgF_2_ layer thickness; and 140 nm of diameter, Figure 9a) and the possible structural changes compared to the base structure adopted to investigate the reason for the 20 nm shorter wavelength shift in Figure 8b.

The structural changes simulated here are as follows: MgF_2_ layer thickness increased by 5 nm (Figure 9b); MgF_2_ layer radius decreased by 5 nm (Figure 9c); and the RI value of MgF_2_ layer decreased by 0.05 (Figure 9d). The absorption spectra of the structures described above are shown in Figure 9e. Compared to the wavelength of the peak at the longer wavelength in (a), the peak shifted to a shorter wavelength by 33, 0.35, and 14 nm for structures (b), (c), and (d), respectively. Among these cases, the peak wavelength change in (c) was much smaller than 20 nm of shift, and considering the 14 nm of shift in (d), the RI value of MgF_2_ layer has to decrease by more than 0.05 (change of RI value in MgF_2_ from 1.38 to 1.33). Therefore, these cases are expected to be impossible. From this demonstration, structure (b), in which the MgF_2_ layer thickness increased by 5 nm, was likely the most reasonable cause of the 20 nm shorter wavelength shift.

From the demonstration of structural change in MIM nanostructures, the response background of the 20 nm shorter wavelength shift is expected to be as follows: 1. The MgF_2_ layer had gaps or pores between deposited particles formed through thermal deposition process; 2. glycerol molecules penetrated into the gaps or pores in the MgF_2_ layer; and 3. the MgF_2_ layers swelled by less than 5 nm.

To investigate whether the penetration of glycerol into MgF_2_ layer may be possible, the density of the MgF_2_ layer fabricated through thermal deposition was optically evaluated. Here, the RI value of the fabricated MgF_2_ layer was compared to that of pure MgF_2_, and the reduction of the RI value was investigated by the presence of gaps or pores inside the fabricated MgF_2_. For the evaluation, a certain thickness of of MgF_2_ layer was deposited on Si substrate in the same condition as the fabrication process of MIM nanostructures, and the cross-sectional SEM image and thin-film interference spectra of the MgF_2_ layer on the Si substrate were measured. Since the MgF_2_ layer is on the high-RI Si substrate, the RI value of fabricated MgF_2_ was calculated by following equations. Here, *n* is the RI value of MgF_2_ layer, *d* is the thickness of MgF_2_ layer, *m* is the order, *λ*_peak_ is the peak wavelength in the thin-film interference spectrum, and *λ*_dip_ is the dip wavelength in the thin-film interference spectrum.
2*nd* = *m**λ*_peak_(3)
2*nd* = (*m* + 1/2)*λ*_dip_(4)

Figure 10a presents a cross-sectional SEM image of the MgF_2_ layer deposited on a Si substrate. From the image, the thickness of MgF_2_ layer *d* was measured to be 289 nm. Figure 10b presents a typical thin-film interference spectrum of the MgF_2_ layer deposited on a Si substrate. From the spectrum, *λ*_peak_ and *λ*_dip_ were measured to be around 790 and 520 nm of wavelength, respectively. Therefore, the order of thin-film interference *m* was determined to be 1. Through observing the thin-film interference spectrum from 26 points on a substrate, *λ*_dip_ was measured to be 522.2 ± 8.5 nm. Based on these results of the SEM image and the thin-film interference spectrum, the RI value of the MgF_2_ layer fabricated through thermal deposition was calculated to be 1.355 ± 0.022, which is 96.6–99.8% of the RI value of pure MgF_2_, 1.38 [46,47]. From this result, the RI value of the deposited MgF_2_ was low compared to pure MgF_2_, and imperceptible gaps or pores were expected to exist.

From these results, the response of the peak at the longer wavelength was expected to be composed of the surrounding RI response and the swelling of the MgF_2_ layer. Considering the MgF_2_ property to adsorb hydroxyl groups, the peak at the longer wavelength showed a larger response in the case when glycerol, which contained more hydroxyl groups than water and IPA, existed in the solution (Water-Gly, IPA-Gly). In contrast, in the cases where the solution was composed of water and IPA with the same number of hydroxyl group, since there was a less significant difference in the interaction between MgF_2_ and the molecules, even with the mixing ratio changed, the peak at the longer wavelength was expected to have just slight change.

From the results in two absorption peaks of the responses of the component to surrounding solutions, MIM nanostructures fabricated in this work were revealed to respond to the surrounding RI value and change in MgF_2_ layer simultaneously by two plasmon modes. We noted especially that the MIM nanostructures fabricated in this work showed completely different wavelength shift behavior in two absorption peaks for solutions with similar RI values, 32 wt.% glycerol aqueous solution (*n* = 1.3690) and 50 wt.% IPA aqueous solution (*n* = 1.3661) (shown in black dashed squares in Figure 8).

## 4. Discussion

The MIM nanostructures were designed to form LSP modes that generate electric fields outside and inside MIM nanostructures, to obtain multiple kinds of information from sample solutions. We successfully designed MIM nanostructures which presented two absorption peaks at the 600–1000 nm wavelength region that were assigned to the LSP modes that generated electric fields in spatially different locations.

Based on the design, MIM nanostructures with diameter of 139.6 ± 2.8 nm and slope of 70° was fabricated. From the optical characterization of the fabricated MIM nanostructures, the absorption peaks were experimentally observed at the 719 and 907 nm wavelengths. Though the peak wavelengths differed to the simulated ones, considering the similarity of the structures and the number of peaks in the wavelength region, the peaks at the 719 and 907 nm wavelengths were expected to be assigned to the LSP modes distributed outside and inside MIM nanostructures, respectively.

From the investigation of response properties of the fabricated MIM nanostructures to the RI valuse and molecular components of surrounding sample solutions, two absorption peaks showed completely different behaviors: a systematic response to the RI value of the sample solution and a component-dependent response. Especially under the conditions of high glycerol concentration, the peak at the longer wavelength largely shifted (20 nm) to a shorter wavelength. From the simulation analysis and evaluation of the filling rate of the deposited MgF_2_ layer, the 20 nm shorter wavelength shift was expected to due to the swelling of MgF_2_ by the penetration of glycerol into the MgF_2_ layer. Furthermore, the fabricated MIM nanostructures showed completely different wavelength shift behavior in two absorption peaks for solutions with similar RI values, 32 wt.% glycerol aqueous solution (*n* = 1.3690) and 50 wt.% IPA aqueous solution (*n* = 1.3661). From these results, the MIM nanostructures fabricated in this work were indicated to respond to the surrounding RI value and sample components independently by two LSP modes.

The results of this study suggest the possibility of simultaneously measuring multiple elements using two or more LSP modes. This technique can be used for the simultaneous measurement of various elements by changing the surface treatment of Au and the material of the insulator layer. For example, the combination of antibody modification on the Au surface and ion-responsive dielectric film may be applied to clarify the correlation between biomolecules and ions released from a single cell.

## Figures and Tables

**Figure 1 micromachines-13-00257-f001:**
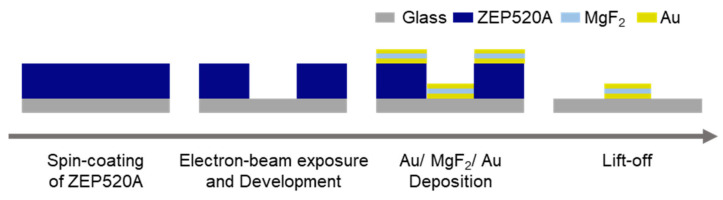
Schematic illustration of the fabrication process of MIM nanostructures.

**Figure 2 micromachines-13-00257-f002:**
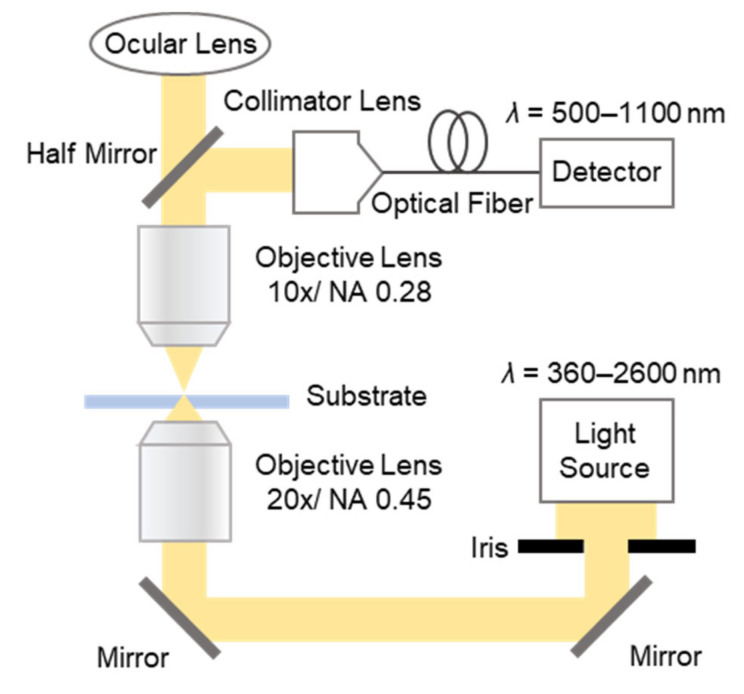
Schematic illustration of the optical micro-spectroscopy system used for the evaluation of the optical characteristics of the fabricated MIM nanostructures.

**Figure 3 micromachines-13-00257-f003:**
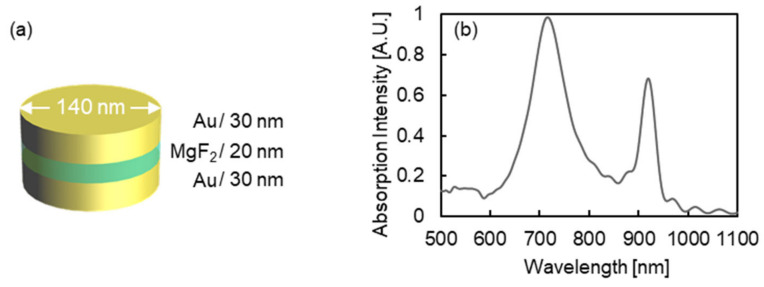
(**a**) Structure and (**b**) absorption spectrum of designed MIM nanostructures. As shown in (**a**), the MIM nanostructures were designed as follows: 30 nm of Au layer thickness; 20 nm of MgF_2_ thickness; and 140 nm of diameter.

**Figure 4 micromachines-13-00257-f004:**
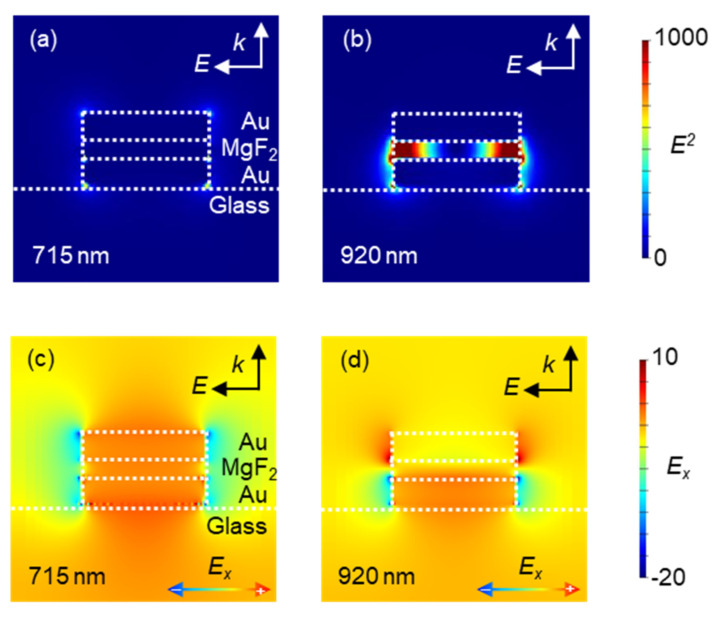
(**a**,**b**) Simulated electric field distributions generated around MIM nanostructures at (**a**) the 715 nm wavelength and (**b**) the 920 nm wavelength. The white arrows in the figures indicate the state of the electric field irradiated to the nanostructure to excite the LSP. (**c**,**d**) Simulated x-vector component of electric field distributions at (**c**) the 715 nm wavelength and (**d**) the 920 nm wavelength. The black arrows in the figures indicate the state of electric field irradiated to the nanostructure to excite the LSP, and the gradient arrows at the bottom of the figures show the correlation between the electric field vector and the color.

**Figure 5 micromachines-13-00257-f005:**
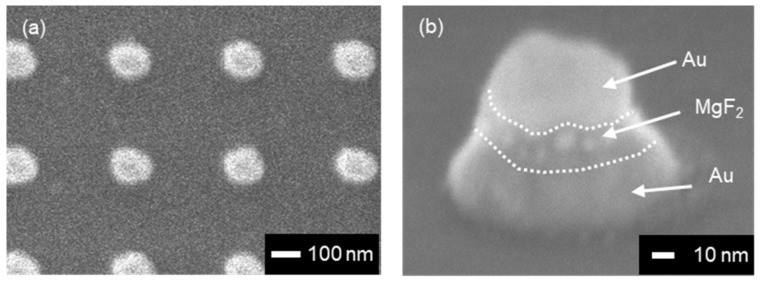
Scanning electron microscopy (SEM) images of the fabricated MIM nanostructures observed from (**a**) the top and (**b**) the side.

**Figure 6 micromachines-13-00257-f006:**
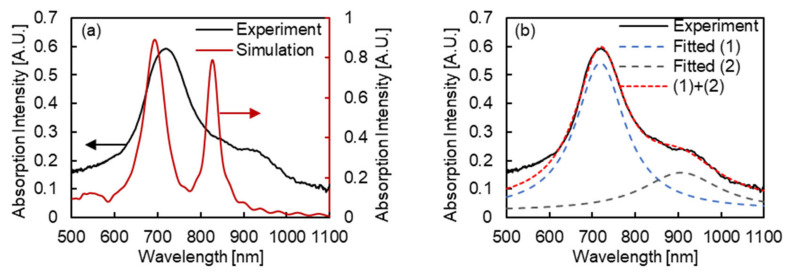
(**a**) Experimental absorption spectrum of fabricated MIM nanostructures and simulated in one of MIM nanostructures imitating the SEM images (140 nm of bottom diameter and 70° slope). The black line indicates the experimental absorption spectrum, and the red line indicates the simulated one. (**b**) Experimental absorption spectrum of fabricated MIM nanostructures and the Lorentz functions fitted to the spectrum. The black line indicates the experimental absorption spectrum, the blue and black dashed lines indicate the fitted Lorentz functions, and the red dashed line indicates the sum of two Lorentz functions.

**Figure 7 micromachines-13-00257-f007:**
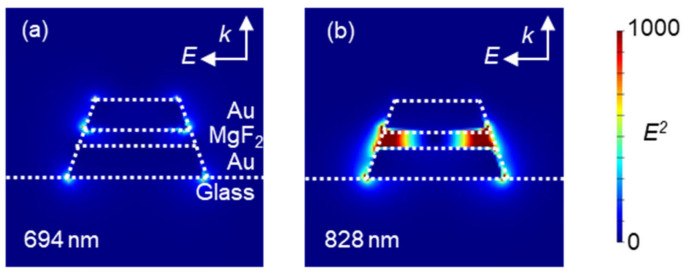
Simulated electric field distribution of MIM nanostructures imitating the SEM images at (**a**) the 694 nm wavelength and (**b**) the 828 nm wavelength. The white arrows in the figures indicate the state of electric field irradiated to the nanostructure to excite the LSP.

**Figure 8 micromachines-13-00257-f008:**
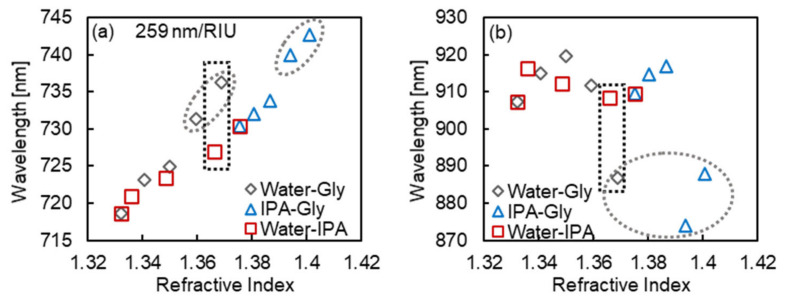
Absorption peak wavelength of the peak at (**a**) the shorter wavelength and (**b**) the longer wavelength of the fabricated MIM nanostructures, depending on the RI values of surrounding sample solutions. The gray hollow diamonds indicate the plots for 0–32 wt.% glycerol aqueous solution (Water-Gly), the blue hollow triangles indicate the plots for 0–32 wt.% glycerol-IPA mixture solution (IPA-Gly), and the red hollow squares indicate the plots for 0–100 wt.% IPA aqueous solution (Water-IPA). The gray dashed ellipses in (**a**) indicate the outliers, which might be attributed to the adsorption of glycerol. The gray dashed ellipse in (**b**) indicates the large shift to the shorter wavelength in the condition of high glycerol concentration. The black dashed squares show the plots for 32 wt.% glycerol aqueous solution (*n* = 1.3690) and 50 wt.% IPA aqueous solution (*n* = 1.3661), in which the two absorption peaks showed completely different responses toward similar RI values of the solution.

**Figure 9 micromachines-13-00257-f009:**
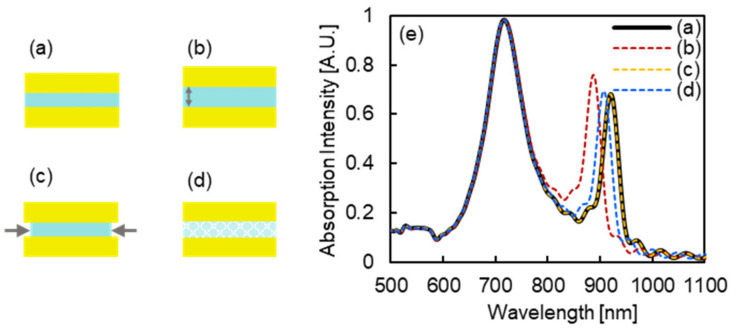
(**a**–**d**) The base MIM nanostructure and the possible structural changes compared to the base structure adopted to investigate the reason for the 20 nm shorter wavelength shift in Figure 8b. (**a**) The base MIM nanostructure with its structural parameters: 30 nm of Au layer thickness; 20 nm of MgF_2_ layer thickness; and 140 nm of diameter. (**b**) An MIM nanostructure in which MgF_2_ layer thickness was increased by 5 nm, (**c**) an MIM nanostructure in which the MgF_2_ layer radius was decreased by 5 nm, and (**d**) an MIM nanostructure in which the RI value of the MgF_2_ layer was decreased by 0.05. (**e**) Absorption spectra of MIM nanostructures in (**a**–**d**). The black line indicates the absorption spectrum of the structure in (**a**), and the red, yellow, blue dashed lines indicate the absorption spectra of the structures in (**b**–**c**), respectively.

**Figure 10 micromachines-13-00257-f010:**
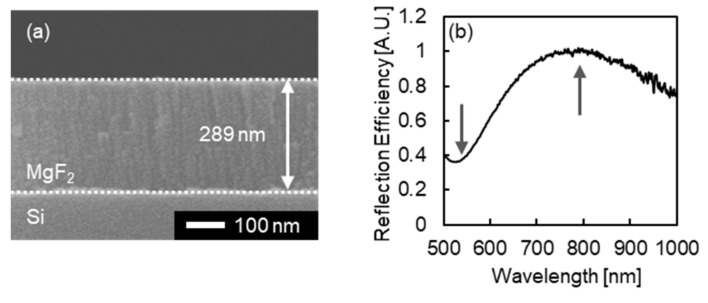
(**a**) Cross-sectional SEM image and (**b**) a typical thin-film interference spectrum of a MgF_2_ layer deposited on a Si substrate. The gray arrows in (**b**) indicate the dip at around the 520 nm wavelength and the peak at around the 790 nm wavelength formed by thin-film interference.

**Table 1 micromachines-13-00257-t001:** Detailed RI values of the sample solutions.

Samples	Glycerol Concentration [wt.%] for Water-Gly and IPA-Gly				
	0	8	16	24	32
	IPA Concentration [wt.%] for Water-IPA				
	0	3.5	18	50	100
Water-Gly	1.3324	1.3407	1.3499	1.3594	1.3690
IPA-Gly	1.3753	1.3805	1.3865	1.3938	1.4009
Water-IPA	1.3324	1.3359	1.3484	1.3661	1.3753

## Data Availability

Data are contained within the article.

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
