# Peer review of "Fabrication of Metal-Insulator-Metal Nanostructures Composed of Au-MgF2-Au and Its Potential in Responding to Two Different Factors in Sample Solutions Using Individual Plasmon Modes"

_micromachines, 2022, doi:10.3390/mi13020257_

Round 1

Reviewer 1 Report

In this paper, a metal insulator metal (MIM) nanostructure was prepared by using magnesium fluoride as insulating layer, and the structure has two absorption peaks in the wavelength range of 500-1100 nm. The structure has two plasmon modes, which is expected to obtain a variety of information of the sample solution. The author needs to solve the following problems before publishing.

  1. The abstract part is too long, and the author needs to reduce his non work introduction.
  2. The “Design of MIM Nanostrctures” part is too simple. The author needs to introduce the parameters of this part in more detail, including the selection of materials and the details of simulation.
  3. The discussion part is too cumbersome, and the author needs to simplify it more.
  4. About “localized surface plasmon”, some relevant literature needs to be mentioned, such as: Phys. Chem. Chem. Phys., 2021, 23 (31), 17041-17048; Physical Chemistry Chemical Physics, 2021, 23, 26864 -26873. About “MIM nanostructures”, some relevant literature needs to be mentioned, such as: Plasmonics 2015, 10, 1537–1543; Plasmonics 2018, 13, 345–352.

Author Response

Dear Reviewer,

Thank you very much for your valuable comments and revisions for our manuscript. Our responses to your comments are described as follows.

The abstract part is too long, and the author needs to reduce his non work introduction.

  • Thank you very much for your valuable advice. We reduced non work introduction of abstract part, instead, the descriptions about the results and conclusion were added in line 16, 18, 19-21, 24-26 and 27-28. The abstract was reduced in total.

The “Design of MIM Nanostrctures” part is too simple. The author needs to introduce the parameters of this part in more detail, including the selection of materials and the details of simulation.

  • Thank you very much for your valuable advice. We have added descriptions in line 116-123, which describe the details of the simulation setup (How the structure was modelled, how the light was introduced, and how the absorption spectrum was obtained) and the setting of optical characteristics of each materials.

The discussion part is too cumbersome, and the author needs to simplify it more.

  • Thank you very much for your valuable advice. We have simplified the discussion part.

About “localized surface plasmon”, some relevant literature needs to be mentioned, such as: Phys. Chem. Chem. Phys., 2021, 23 (31), 17041-17048; Physical Chemistry Chemical Physics, 2021, 23, 26864 -26873. About “MIM nanostructures”, some relevant literature needs to be mentioned, such as: Plasmonics 2015, 10, 1537–1543; Plasmonics 2018, 13, 345–352.

  • Thank you very much for your valuable advice. We have added the relevant information. About “localized surface plasmon”, we added references No. 5,6 and also added a sentence to mention it in line 33. About “MIM nanostructures”, we added references No. 32,33 and also added a sentence to mention it in line 60.

Based on your valuable advice, we have revised our manuscript.

Reviewer 2 Report

The manuscript concerns the investigations of a metal–insulator–metal LSP nanostructures designed to reveal two plasmon modes independently responding to the changes in surrounding medium and inside insulator layer.

The subject matter appears to be interesting for a reader of the journal. The manuscript covers the latest research in modeling of LSP structures. In particular, the authors tried to demonstrate a linear response of one plasmon mode toward the RI of surrounding medium and the shift of the other plasmon mode under the conditions where glycerol is present at high concentration.

The title of the paper is accurate and clearly identifies the subject matter. The manuscript is formatted according to the journal style guide. The manuscript is clearly written and logically organized. The work is placed in proper context and a related work is adequately referenced.

The structural design of a “regular shape” (Fig. 3a) and a “tapered shape” (Fig. 5) structure was evaluated using optical simulation software. Electric field distributions and absorption spectra were obtained for these structures.

The drawback of the work is the significant difference between the experimental and simulated absorption spectra of the produced nanostructure (Fig. 6a). The difference probably results from a significant difference in the shape of the simulated structure (Fig. 7) in comparison to the actual structure (Fig. 5b) and different values of material parameters in relation to the designed values. In order to interpret the results, the authors used two Lorentz functions adjusted to the spectrum and estimated the influence of geometrical and material parameters of the layers on changes in the spectrum of the system as a result of changes in the parameters of the surrounding environment. Interpretation of the results seems reasonable, despite significant differences between simulation and experiment. In this context, it seems that the difference in the response of the system to changes in the refractive index of the surrounding environment and to changes in the composition of the solution by changing the position of the two peaks of the spectrum of the system has been demonstrated clearly enough. Due to the fact that this type of MIM system is being analyzed for the first time in the literature, it seems worth showing the results despite the above-described drawbacks. Nevertheless, a few issues should be addressed:

  • the first half of the Abstract section should rather be placed in the Introduction section; the Abstract section should briefly present the results and the final conclusion;
  • information on the assumed diameter of the structure should be included in chapter 2.2.2;
  • numerical values of material parameters used in modeling should be given in chapter 3.1.

Author Response

Dear Reviewer,

Thank you very much for your valuable comments and revisions for our manuscript. According to your comment, we added the description about the cause of difference between the simulation and experimental spectra from the viewpoint of material in line 246-247. Our responses to your comments are described as follows.

The first half of the Abstract section should rather be placed in the Introduction section; the Abstract section should briefly present the results and the final conclusion.

  • Thank you very much for your valuable advice. We reduced first half of abstract, and added the descriptions about results and conclusion in line 16, 18, 19-21, 24-26 and 27-28.

Information on the assumed diameter of the structure should be included in chapter 2.2.2.

  • Thank you very much for your valuable advice. We have added descriptions about assumed diameter of MIM nanostructure to be fabricated in line 129-130.

Numerical values of material parameters used in modeling should be given in chapter 3.1.

  • Thank you very much for your valuable advice. We have added descriptions about the material parameters in line 116-123. Since the material parameters were set fitting to the database included in the simulation software through the simulation setup procedure, we added the sentence in chapter 2.2.1 instead of chapter 3.1.

Based on your valuable advice, we have revised our manuscript.

Round 2

Reviewer 1 Report

The article has been modified by the system and can be received.